# Changes of Lake Area, Groundwater Level and Vegetation under the Influence of Ecological Water Conveyance—A Case Study of the Tail Lake of Tarim River in China

**Xinfeng Zhao** [1,2,3], **Hailiang Xu** [1,2,*] **and Peng Zhang** [1,2]

1   State Key Laboratory of Desert and Oasis Ecology, Xinjiang Institute of Ecology and Geography, Chinese Academy of Sciences, Urumqi 830011, China; zxinfeng_xjbg@126.com (X.Z.); zhangpeng_xjbg@126.com (P.Z.)
2   Xinjiang Aksu Oasis Agro-Ecosystem Observation and Experiment Station, Aksu 830011, China
3   University of Chinese Academy of Sciences, Beijing 100049, China
*   Correspondence: xulh@ms.xjb.ac.cn

**Abstract:** To study the changes of water and vegetation coverage, groundwater level and plant diversity of lakes at the end of Tarim River in Northwest China, the changes of various indicators in more than 20 years (2000–2019) were analyzed through field investigation and indoor remote sensing methods. The results showed that (1): with the initiation of the development of ecological water conveyance project, water and vegetation areas increased significantly, especially the trend of vegetation areas becoming more significant, and area of sandy land decreased significantly. (2): the plant diversity increased in the early stage of ecological water conveyance, however, with the increase of lake area and groundwater level, the species composition tends to be simplified. According to the variation characteristics of species importance value in the overflow area in recent 20 years, it is found that the top communities of plant succession are *Phragmites australis* and *Hexinia polydichotoma*. (3): with the increase in the lake area, the groundwater level showed an up-lifted trend, the correlation between the two was significant, but there was a lag in the response of the groundwater level. (4): The intra-annual variations in the lake areas were considerable before and after the ecological water conveyance.

**Keywords:** lake area; vegetation area; groundwater level; plant diversity





## 1. Introduction

Lakes, as an important carrier of water resources in arid areas, maintain the balance of fragile ecosystems and the needs of human economic and social development. They are an important part of water cycle in arid ecosystems [1,2]. Inland lakes, especially the tail lakes, are in a relatively independent inland humidity cycle system, and are affected by climate change and human activities [3,4]. The disappearance or shrinkage of lakes not only leads to the desertification of surrounding oasis wetlands, but also the exposed dry lake bottom sediments become the source of wind erosion and transportation, resulting in frequent saline alkali dust storms, which poses a serious threat to the production, life and the health of residents in the basin [5]. Additionally, with consequences for nutrients' internal load in the lakes, for fish species and, eventually, fisheries, for migratory birds and others [2,3].

The research on the temporal and spatial changes of tail lakes in a long time series has important practical significance for lake early warnings, water resources management and sustainable development in arid areas. The changes of these tail lakes are not only affected by regional climate change, but also vulnerable to human activities such as irrigation and water diversion. Human disturbance caused by population growth, socio-economic development and highly intensive use of resources is directly or indirectly leading to the degradation of ecosystem and ecological environment, such as the reduction of ecosystem productivity, the reduction or loss of biodiversity, the reduction of lake area and the increase

of climatic disasters, which seriously threaten the human sustainable development. The excessive consumption of surface water resources by agricultural irrigation has gradually become the main contradiction restricting the social leapfrog development of Tarim River Basin in China [6].

The study on the succession law of plants and the spatial distribution characteristics of vegetation in the periphery of the tail lake is conducive to the renewal of biological resources and the sustainable development of local ecological environment [7,8]. It has important theoretical and practical significance for vegetation restoration, soil and water conservation and biodiversity protection in arid desert areas [2,9]. At the same time, as a precious water resource in arid areas, lakes are important for the growth and development of surrounding plants succession change and ecosystem stability play a more important role.

Influenced by the global trend, governments are aware of the seriousness of ecological degradation and the urgency of carrying out ecological restoration [10,11]. Some ecological restoration programs are implemented to mitigate ecological degradation, e.g., the "Reforestation of Fragmented Forests Project" in Kianjavato, Madagascar [12], the "Megadiversity Atlantic Forest Biome Restoration Project" in Iracemápolis, Brazil [13], and the "Comprehensive Management" in the Tarim River Basin [14]. From the 1970s to 2000s, there were 320 km of river has been cut off in the lower reaches of the Tarim River because of large-scale reclamation of arable land in the source and in the upper and middle reaches. The Chinese government has invested 10.7 billion RMB on the "Ecological Water Conveyance Project" (EWCP) since 2000. Tarim River Basin transfers ecological water to Taitema Lake, the tail lake of Tarim River Basin, through water resource conservation and management of the whole basin and water transfer from Bosten Lake. The increasingly serious ecological degradation in the lower reaches of Tarim River has drawn the attention of the Chinese government and all sectors of society. The river finally arrived at the tail lake in 2003, the hydrological process integrity of Tarim River was restored, and a certain water surface was formed.

The study of inland river tail Lake in geography is mainly carried out from two angles. First, from the perspective of regional geography, study the geographical location, spatial scope, geomorphic characteristics, hydrological correlation, vegetation ecology and other issues of tail-lake. Some studies take the basin as a unit [15,16]. Second, from the perspective of historical geography, take the lake as the indicator of basin environmental change, and explore the relationship between the environmental change of tail-lake and river diversion, oasis change, historical reclamation and population change in a certain time series [17,18].

As for the restoration and reconstruction of lakes in arid areas in the world, the Aral Sea Basin in Central Asia is the most prominent and attracting the most attention, which is also very similar to the vast lakes in arid areas in China. The restoration of the Aral Sea involves many countries, mainly in the cross-border cooperation of water and some engineering measures [19]. For example, Kazakhstan has built a dam between the large and small Aral seas to prevent the flow of water from the small Aral Sea into the large Aral Sea, so as to maintain its water area. The researchers put forward three schemes of water diversion for the surface restoration of the Aral Sea, all of which are cross basin long-distance water transmission [20]. Additionally, the development and management of water resources in the Colorado River Basin in the United States is based on the annual runoff of the upper and lower reaches of the Colorado River Basin [21]. Finally, it is considered that one of the main means of rational allocation of river basin water resources is to formulate River Basin Water Diversion Scheme [22–24].

Chen et al. [25] analyzed the environment around Taitema Lake from 1973 to 2006, and Abdumijiti [26] and Zhang et al. [27] pointed out that even in 2004, when it formed an area of about 300 km$^2$, the Taitema Lake had still been in a state of biodiversity loss and extreme desertification based on remote sensing data from 1973 to 2014 [14]. Huo et al. [28] and Wang et al. [29] discussed the water area change of Taitema Lake and its influence on surrounding animals and climate environment in 2020.

So far, the EWCP has been carried out for more than 20 years, and the largest area of Taitema Lake reached 511 km$^2$ in 2017. In the years of 2017 and 2018, the water conservancy department of the Chinese government put forward an urgent question to us: "How large should the suitable water area of the terminal lake be maintained? Is it absolutely certain that the bigger the better?" To meet the objectives and clarify a hypothesis, i.e., larger lake areas result in larger values of diversity and species richness.

Additionally, no research has been carried out on the vegetation dynamic and the response characteristics of groundwater level, discussed the internal mechanism of surface vegetation change and the reasons and internal mechanisms of the changes. This study, therefore, was conducted to partly fill in this information gap with the help of remote sensing image interpretation data and ground vegetation survey data, hoping to provide technical and theoretical support for the related studies in other arid regions.

## 2. Materials and Methods

### 2.1. Overview of the Research Area

Taitema Lake is a low-level alluvial plain in Tarim Basin, China. The location of the study area is shown in Figure 1. At present, it is located at the intersection of Tarim River and Cherchen River (these two rivers are two of the ten rivers that constitute the "nine sources and one main stream" of Tarim River). It belongs to Ruoqiang County in China (the largest county in China), with an annual precipitation of 28.5 mm and an annual potential evaporation of 2920.2 mm. The main natural vegetation is sparse desert arid halophyte.

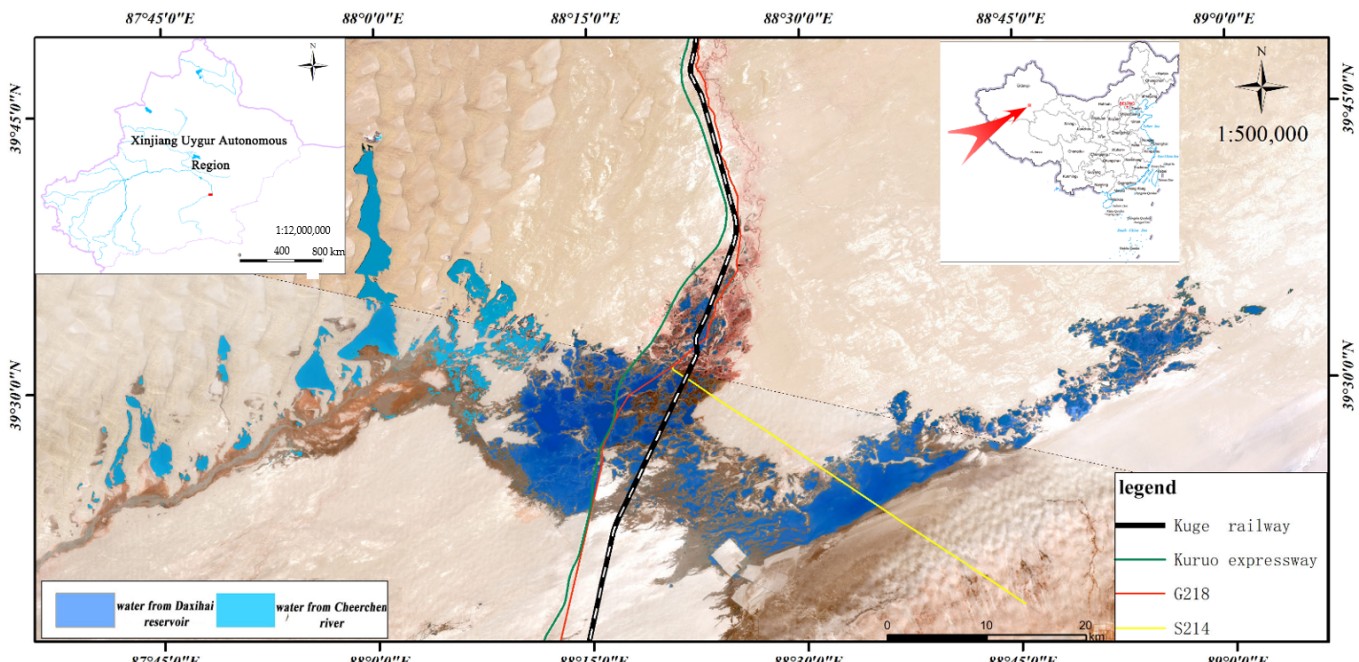

**Figure 1.** Schematic diagram of the location of Taitema Lake.

Rivers always flow from high-altitude areas to low-altitude areas. Due to the climate change, river changes and diversion, Taitema Lake gradually dried up in the 1970s, forming a situation of serious desertification. Fluvial and lacustrine sediments are widely distributed in the study area. The particle composition is mainly powder and fine sand, and the structure is loose. When the water conditions change or the vegetation is damaged, it is most easy to sand locally under the strong wind to form various types of sand dunes. At the same time, the moving speed of sand dunes is also very fast, and the moving of sand dunes often blocks the traffic. From 2000 to now, the ecological water transportation project has been carried out for 20 years.

### 2.2. Sample Plot Distribution and Vegetation Investigation

2.2.1. Sample Plot Distribution

Through years of field investigation, primary monitoring data of Taitema Lake from 2000 to 2020 were collected. In the present study, over-flow plots and non-overflow plots (i.e., control plots) were set up in each sampling zone and field investigations and data collection were carried out to investigate vegetation change under flooding and non-flooding conditions. The choice of sampling plots was influenced by the road network proximity, and the final sample plots in Table 1 and Figure 2 were determined after multi-directional layout and multiple screening during the selection of sample plots.

**Table 1.** Geographical location of the monitoring plots.

| | Plot | Size of Plot | Longitude | Latitude | Altitude (m) |
|---|---|---|---|---|---|
| | Plot 1 | 50 × 50 m | E88°32′42.0″ | N37°34′7.9″ | 798.2 |
| | Plot 2 | 50 × 50 m | E88°22′23.9″ | N39°32′16.5″ | 793.4 |
| | Plot 3 | 50 × 50 m | E88°21′22.8″ | N39°31′30.3″ | 793.3 |
| | Plot 4 | 50 × 50 m | E88°17′03.0″ | N39°28′55.8″ | 796 |
| | Plot 5 | 50 × 50 m | E88°15′43.4″ | N39°26′33.6″ | 799.5 |
| | Plot 6 | 50 × 50 m | E88°11′53.5″ | N39°14′6.3″ | 801 |
| | Plot 7 | 50 × 50 m | E88°23′23.7″ | N39°29′40.2″ | 802 |
| | Plot 8 | 50 × 50 m | E88°22′51.5″ | N39°29′56.7″ | 791 |
| | Plot 9 | 50 × 50 m | E88°22′23.1″ | N39°30′17″ | 786 |
| Flooding area | Plot 10 | 50 × 50 m | E88°22′15.9″ | N39°30′14.7″ | 794 |
| | Plot 11 | 50 × 50 m | E88°22′06.8″ | N39°30′19.7″ | 794 |
| | Plot 12 | 50 × 50 m | E88°21′56.3″ | N39°30′15.9″ | 794 |
| | Plot 13 | 50 × 50 m | E88°23′41.3″ | N39°29′41.1″ | 791 |
| | Plot 14 | 50 × 50 m | E88°23′53.5″ | N39°29′34.7″ | 794 |
| | Plot 15 | 50 × 50 m | E88°24′12.7″ | N39°29′24.6″ | 795 |
| | Plot 16 | 50 × 50 m | E88°21′30.2″ | N39°30′15.8″ | 806.4 |
| | Plot 17 | 50 × 50 m | E88°20′45.5″ | N39°30′9.6″ | 806.5 |
| | Plot 18 | 50 × 50 m | E88°21′46.0″ | N39°30′14.4″ | 807.5 |
| | Plot 19 | 50 × 50 m | E88°16′28.1″ | N39°26′37.7″ | 795.3 |
| | Plot 20 | 50 × 50 m | E88°16′13.5″ | N39°26′36.3″ | 798.5 |
| | Plot 21 | 50 × 50 m | E88°21′23.0″ | N39°31′18.6″ | 808.9 |
| | Plot 1 | 50 × 50 m | E88°20′41.0″ | N39°30′59.0″ | 807.5 |
| | Plot 2 | 50 × 50 m | E88°20′45.9″ | N39°30′54.0″ | 807.9 |
| | Plot 3 | 50 × 50 m | E88°20′56.3″ | N39°30′51.4″ | 807.8 |
| | Plot 4 | 50 × 50 m | E88°21′56.8″ | N39°30′12.2″ | 807.7 |
| | Plot 5 | 50 × 50 m | E88°23′28.6″ | N39°29′41.2″ | 808 |
| | Plot 6 | 50 × 50 m | E88°23′43.4″ | N39°29′50.6″ | 807.9 |
| | Plot 7 | 50 × 50 m | E88°23′39.8″ | N39°30′7.9″ | 807.5 |
| | Plot 8 | 50 × 50 m | E88°23′25.8″ | N39°29′40.9″ | 807.9 |
| Groundwater elevated area | Plot 9 | 50 × 50 m | E88°23′15.7″ | N39°29′24.3″ | 806.5 |
| | Plot 10 | 50 × 50 m | E88°23′27.6″ | N39°29′26.5″ | 807.1 |
| | Plot 11 | 50 × 50 m | E88°25′20.6″ | N39°36′23.8″ | 812.7 |
| | Plot 12 | 50 × 50 m | E88°25′20.6″ | N39°36′24.1″ | 811.8 |
| | Plot 13 | 50 × 50 m | E88°9′22.7″ | N39°26′36.9″ | 810.6 |
| | Plot 14 | 50 × 50 m | E88°11′13.9″ | N39°17′20.4″ | 812.1 |
| | Plot 15 | 50 × 50 m | E88°12′25.9″ | N39°16′50.5″ | 810.8 |
| | Plot 16 | 50 × 50 m | E88°12′10.4″ | N39°15′45.3″ | 812.1 |
| | Plot 17 | 50 × 50 m | E88°14′8.1″ | N39°18′46.0″ | 810.4 |
| | Plot 18 | 50 × 50 m | E88°15′22.3″ | N39°19′28.2″ | 809.5 |

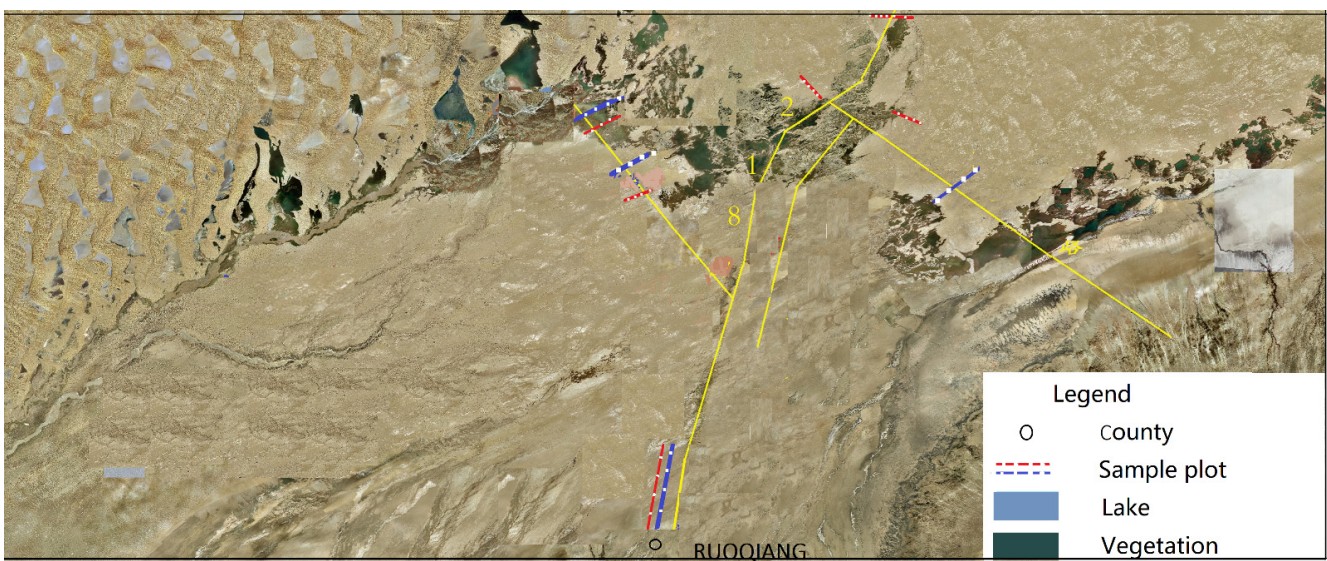

**Figure 2.** Geographical location of the monitoring plots.

Among the 39 fixed monitored plots, 21 were in flooding and 18 were in groundwater elevated areas. In 2009, eight plots have been flooded for prolonged periods and were no longer used as fixed monitoring plots. Therefore, only 31 plots remained by 2019. The groundwater elevated area was located at the end of the sampling transects and was relatively far from the lakeside. The groundwater elevated area were 1–5 km away from the shoreline.

### 2.2.2. Vegetation Sampling

In each sampling plot of 50 m × 50 m that was set up permanently, total vegetation cover, species and the vegetation cover by individual species were investigated. In addition, three small quadrats measuring 1 m × 1 m were set up permanently in each of the 50 m × 50 m plots using the diagonal method. In addition, another three plots of 1 m × 1 m in 50 m × 50 m plots were randomly selected in order to increase the repetition; in each quadrat, plant species composition and abundance and vegetation coverage were measured.

### 2.3. *Measurement of Vegetation Area in Taitema Lake Zone*

Based on the MODIS-EVI data of Taitema Lake area from 2000 to 2019, the temporal and spatial variation characteristics of the coverage area of Taitema Lake area are clarified to evaluate the effect of ecological water delivery on vegetation restoration. First of all, with the help of ENVI 5.0 software platform, the spatial distribution range of vegetation cover is defined year by year, and then the spatial distribution data of vegetation in Taitema Lake are extracted using ArcGIS 10.8 spatial analysis function. The vegetation index data used for vegetation cover extraction in the present study were Enhanced Vegetation Index (EVI) data of the MOD13Q1 product with a spatial resolution of 250 m and a January 2001 to December 2019 time series.

Pre-processing the EVI data of the remote sensing data occurred via processes such as format conversion, Mosaic, projection, projection transformation and extraction of the study area. To minimize noise, the data images were processed further using Savitzky–Golay filtering and maximum value composite synthesis processing of remote sensing data.

There are 23 EVI data in a year (one every 16 days). The maximum synthesis method [28] is used to extract the maximum EVI data of the observation pixels from the 23 images to generate new EVI data. Therefore, the obtained data represent the EVI data under the annual optimal vegetation growth conditions. Therefore, the annual EVI data representing the optimal vegetation growth conditions were obtained.

The pixel dichotomy model is an effective method for vegetation cover inversion. The formula is as follows:

$$F_c = \frac{EVI - EVI_{soil}}{EVI - EVI_{veg}} \tag{1}$$

where $F_c$ is the vegetation cover, which is represented by two decimal numbers in the present study. $EVI_{soil}$ is the EVI value of pure bare soil pixels in the study area, and $EVI_{veg}$ is the EVI value of pure vegetation pixels. In the present study, the EVI values at 5% and 95% of the histogram of EVI images in the study area were considered to represent $EVI_{soil}$ values and $EVI_{veg}$ values, respectively.

### 2.4. Interannual and Intra-Annual Data of Lake Area

(1)   Interannual Maximum Lake Area Date

In this paper, the maximum lake area data of each year from 1998 to 2014 were derived from the study of Abdumijiti [26] and the study of Chen [25]. The area data after 2014 were obtained by ourselves through downloading images and extracting data.

(2)   Intra-Annual Measurement of Lake Area

Data of intra-annual area in 2016, 2017, 2018 and 2019 were measured by ourselves through downloading images and extracting data. The area extraction method was the same as that of Abdumijiti [26] and Chen [25].

### 2.5. Groundwater Level Date

Groundwater level data include: (1) groundwater level data in the lake area, (2) groundwater level data from the outer edge of the lake area, and (3) groundwater level data of four sections 40 km, 120 km, 150 km and 170 km away from the Lake Center (these four sections belong to the lower reaches of Tarim River and are perpendicular to the river channel of Tarim River). In the data adopted in this paper, some groundwater wells are 50 m away from the river channel and some are 1050 m away from the river channel.

### 2.6. Vegetation Area under Different Vegetation Coverage Gradients

Figure 3 shows the composition of vegetation area of different coverage gradients. Vegetation coverage was divided into 6 different gradients, namely, (1) sandy land (vegetation coverage < 10%), (2) low coverage (10–20% vegetation coverage), (3) medium and low coverage (vegetation coverage is 20–40%), (4) Medium coverage (vegetation coverage is 40–60%), (5) medium and high coverage (vegetation coverage is 60–80%), And (6) high coverage (vegetation coverage > 80%). It can be seen from Figure 3 that the vegetation area of the low coverage level accounts for the largest proportion, and the vegetation area of other levels accounts for a relatively small proportion. Therefore, when analyzing the change of vegetation area in this study, only the data of total vegetation area and vegetation area of low coverage level were used.

As Taitema Lake itself is a flat, low-altitude area, and the elevation difference (altitude) of the whole area is no more than 10 m, the vegetation (mainly *Phragmites australis*, whose important value accounts for more than 0.98) close to the center of the lake and river course with relatively deep water depth always maintains good growth and large vegetation coverage. In areas far away from the center of the lake and river, the vegetation (mainly *P. australis*, whose important value accounts for more than 0.80) has been low or even disappeared due to the lack of normal water supply. Therefore, this part of vegetation is always preserved no matter if in the year when the lake surface is large in wet season or in the year when the lake surface is small in dry season. The area far away from the lake center and river channel must not be retained in dry years, so the change of vegetation area of the tail lake is mainly the change of low coverage vegetation area. It can also be seen from Figure 3 that the vegetation area of low coverage level accounts for the largest proportion, and the vegetation area of other levels accounts for a very small proportion.

Therefore, when analyzing the change of vegetation area in this study, only the vegetation area data of total vegetation area and low coverage level are used.

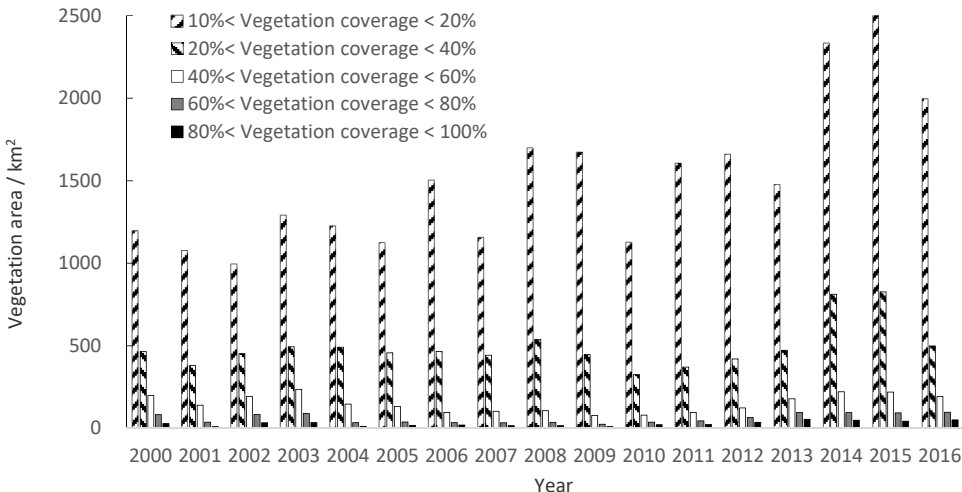

**Figure 3.** Vegetation area under different vegetation coverage gradients.

### 2.7. Statistical Analysis

Determination of important value:

Important value can reflect the dominant species in the community and objectively express the status and role of each species in the community. Its calculation method [3] is as follows:

$$\text{Important value} = (\text{relative density relative coverage relative frequency})/3 \tag{2}$$

Relative density = number of plants of a species/sum of the number of plants of all species × 100%

Relative coverage = coverage of a species/coverage of all species and × 100%

Relative frequency = frequency of a species/sum of frequencies of all species × 100%

We account for the number of species, calculated the species frequency and the Pielou evenness index in each quadrant.

The trend changes for the plant diversity index, vegetation area and lake area were analyzed using the Mann–Kendall monotony trend test. In the test process, when the statistic (i.e., Zc) is positive, it reflects an upward trend; whereas, when Zc is negative, it reflects a downward trend. If the checking standards are salient ($|Zc| \geq |Z_{0.05}| = 1.96$ at the 95% level, $|Zc| \geq |Z_{0.01}| = 2.58$ at the 99% level), then the trend is credible; otherwise, it is not.

Simpson index ($D_1$) [30]:

$$D_1 = 1 - \sum P_i^2 \tag{3}$$

Species richness ($R$):

$$R = S \tag{4}$$

Shannon-Wiener index [31]:

$$H = -\sum_{i=1}^{S} (P_i \ln P_i) \tag{5}$$

Pielou evenness index (*Ep*):

$$Ep = -\sum_{i=1}^{S} (P_i \ln P_i) / \ln S \qquad (6)$$

$P_i$ is the relative importance of species *i* (calculated using frequency). *S* is the total number of species in the quadrat.

Relative importance of a species is calculated as follows:

$$P_i = W_i / W \qquad (7)$$

$W_i$ is the individual number of species *i*; *W* is the sum of all individuals from different species in the quadrat.

## 3. Results

### 3.1. Variation in Lake Area and Response of Groundwater Level

With the implementation of the ecological water conveyance project, the lake area has increased from 58.79 km$^2$ in 2000 to 340 km$^2$ in 2020, and reached 511 km$^2$ in 2017. Among them, from 2000 to 2006, the lake area generally showed an upward trend. From 2007 to 2009, the lake area decreased due to dry water. From 2010 to 2018, the lake area generally showed an upward trend due to more wet years. In 2019, there was less water and the lake area became smaller, as shown in Figure 4. During the whole period, the change of lake area showed a significant increase trend, which showed that the response of lake area to ecological water conveyance continued to expand.

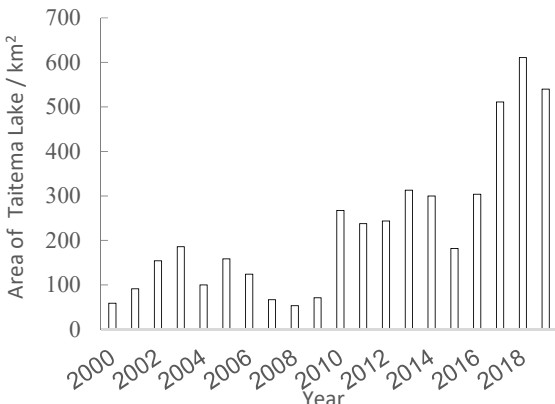

**Figure 4.** Variation in the area of Taitema Lake.

The Taitema Lake formed a lake surface of about 180 km$^2$ in 2003, which also ended the history of the disconnection of the lower reaches of the Tarim River. Later, the ecological transmission has gradually normalized. Due to the water transmission effect ensured by a variety of water transmission means, the area of Taitema Lake has increased in fluctuation and stabilized at more than 200 km$^2$ since 2010. With the progress of ecological water conveyance and the increase of lake area, the groundwater level has increased significantly compared with that before water conveyance.

Figure 5A shows the groundwater level of two monitoring wells in Taitema Lake from 2009 to 2011. The shallowest groundwater level in 2011 was within 0.5 m, 88.9% shallower than the average groundwater level in 2009 (4.5 m), indicating that the groundwater level around lake is becoming shallower. Figure 5B shows the groundwater level of the sections 170 km, 150 km, 120 km and 40 km away from Taitema Lake. Compared with 2000, the groundwater level of the above sections at different distances from the lake has increased significantly: in 2018, the groundwater level of the sections 170 km, 150 km, 120 km and 40 km away from Taitema Lake increased by 2.28, 3.31, 5.23 and 5.05 m, respectively (the

vertical distance between each section and Tarim River is 50 m), and the groundwater level of the above sections increased by 0.52 m, 0.65 m and 1.28 m and 1.65 m (1050 m perpendicular to Tarim River).

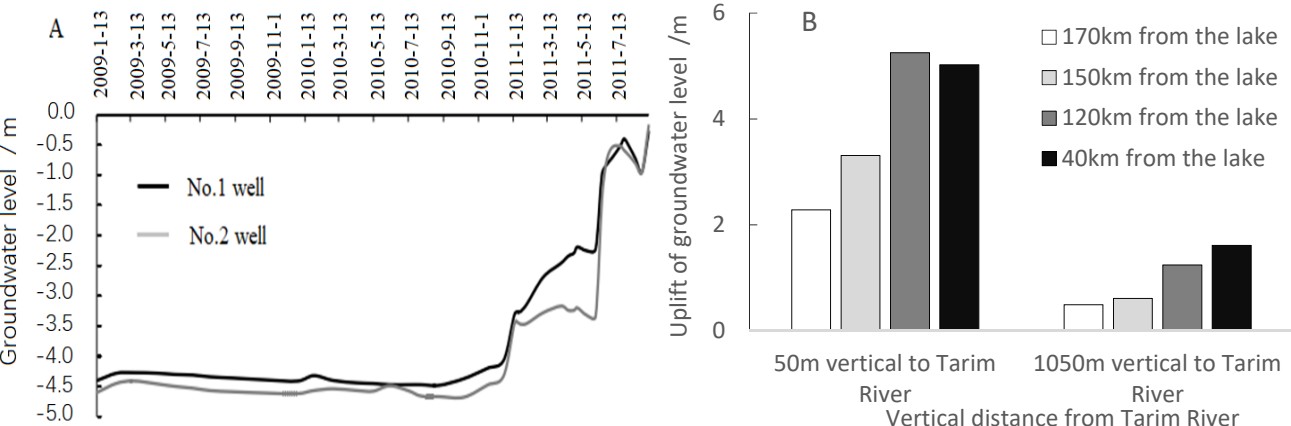

**Figure 5.** Groundwater level change around Taitema Lake (**A**) and at different distances from the lake (**B**).

With the increase in the lake area, the surrounding groundwater level also rose, but there was a lag phenomenon in the early stage: during the period of 2000 to 2010, there was little change in groundwater level (Figure 6), and it gradually increased as time went on by the end of 2010. This is also highlighted in Figure 5A above. This is mainly because the tail lake is in the condition of long-term drought, and the ecological environment is extremely bad, so it cannot be effectively restored in the short term.

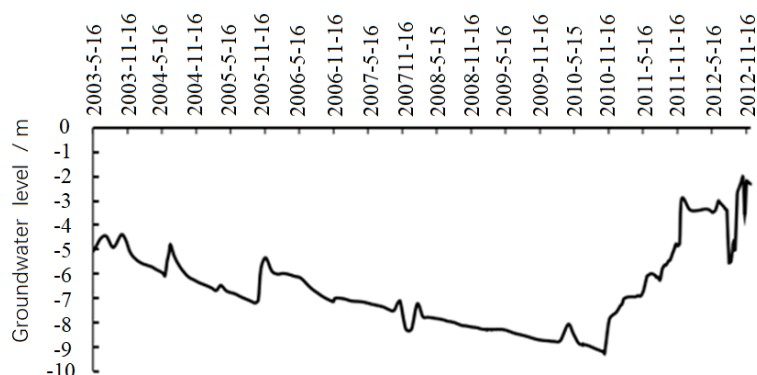

**Figure 6.** Groundwater level changes of Kogan No. 1 monitoring well.

From 2002 to 2010, the groundwater level around Taitema Lake remained at about 7.2 m, that is, the groundwater level around Taitema Lake can be responded to only after 10 years of ecological water conveyance, which reflects the lag of groundwater response to ecological water conveyance (Figure 6).

From the above analysis, it can be seen that with the increase of lake area, the groundwater level around Taitema Lake has also been improved. Is there a certain relationship between Lake area and groundwater level? Due to the short monitoring time series of the two groundwater levels in the Taitema Lake area, we selected the groundwater level data of two monitoring wells (J3 and J4) in the Kaogan section (originally the groundwater level monitoring scope of the lower reaches of the Tarim River), which is the closest monitoring section to the Taitema Lake, to establish a relationship with the lake area, as shown in Figure 7. The results show that the Pearson correlation coefficients between the groundwater level and the lake area of J3 (Figure 7A) and J4 (Figure 7B) are −0.79 and

−0.70, respectively ($p < 0.01$), indicating that there is a very significant negative correlation between the groundwater level and the lake area, that is, the larger the lake area, the shallower the groundwater depth.

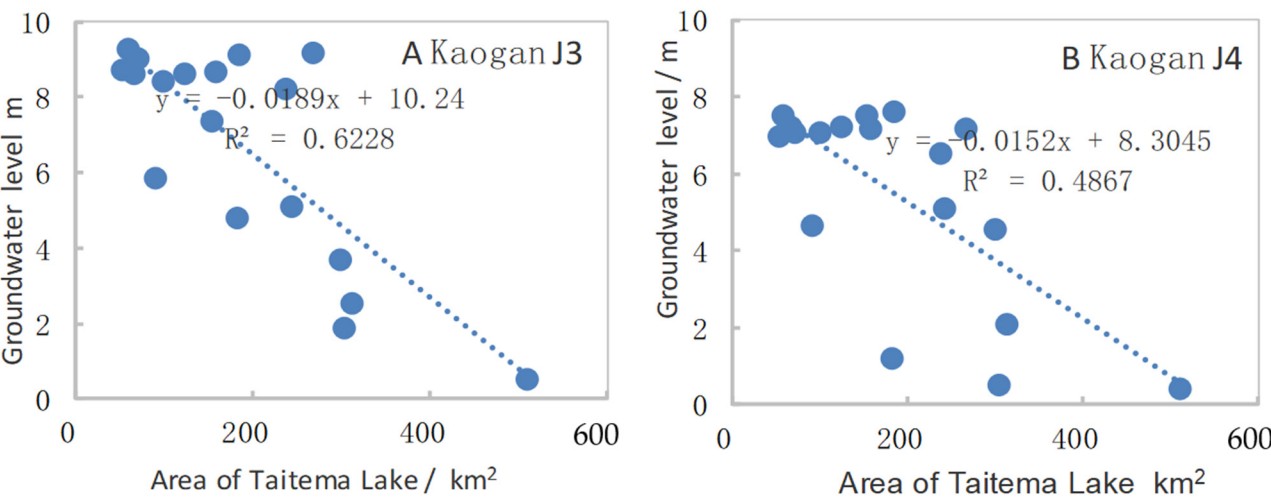

**Figure 7.** Relationship of groundwater of Kaogan section and area of Taitema Lake.

### 3.2. Variation in Vegetation Coverage

After ecological water conveyance, the groundwater depth around Taitema Lake is rising, which also has a certain impact on the growth of vegetation around the lake. As shown in Figure 8, from 2000 to 2016, the vegetation coverage area under each gradient level is increasing. Among them, from 2000 to 2010, this improved effect was not obvious. After 2010, the groundwater has been gradually improved after long-term recharge, and the improvement effect has become more and more significant in the following years. The vegetation coverage area around the lake increased, and area of sandy land decreased significantly. Both the vegetation area with low coverage (coverage 10–20%) (Figure 8A) and the vegetation area of all grades (10–20%, 60–100%, 40–60%, 60–100%) (Figure 8B) showed an increasing trend.

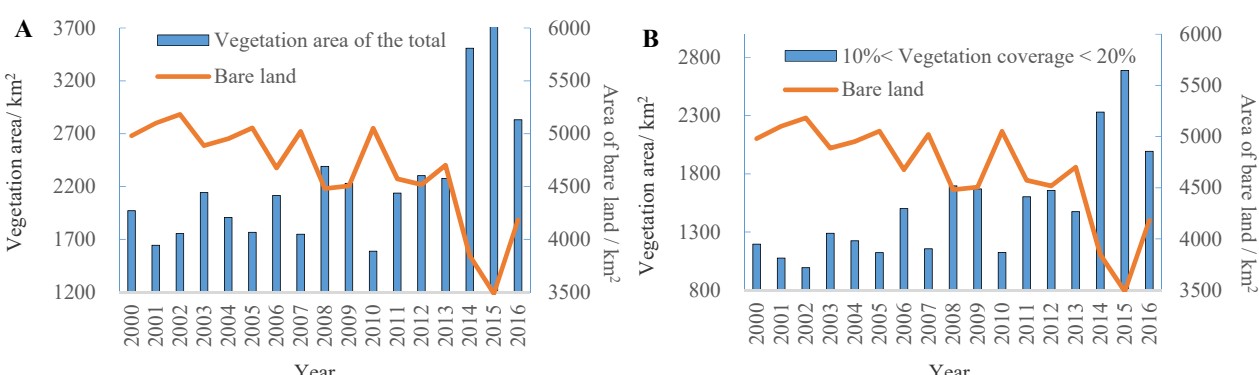

**Figure 8.** Vegetation coverage (**A**) and vegetation area (**B**) of the terminal of the Tarim River.

Mainly due to the long period of river cut-off (nearly 30 years), the sharp decrease in the vegetation quantity and the expansion of the sand area under conditions of sandstorm climate, the vegetation coverage of Taitema Lake seldom changed during the early period (2000–2003). After the EWCP had been implemented 20 years, it was found that, with the increase in water inflow, the vegetation area was expanding. It could be seen that water was beneficial to the growth of plants, and the water brought by ecological water delivery has a positive impact on the growth and recovery of plants in Taitema Lake.

During the past 20 years of ecological water transportation, the vegetation around Taitema Lake showed different characteristics at different water transportation stages. From 2000 to 2016, the vegetation coverage area of Taitema Lake area showed an increasing trend (Figure 8), indicating that the vegetation area was very sensitive to the change of water area, and indicating the vegetation growth recovery after the implementation of the EWCP. However, due to the dry year from 2008 to 2010, the vegetation area decreased significantly from 2008 to 2011. From 2011 to 2016, the amount of ecological water entering the lake increased significantly, and the vegetation area in the whole study area showed a general growth trend, but did not show a sustained growth. The main reason may be that from 2011 to 2016, with the increase of lake area and water depth, some vegetation was submerged below the water surface. During remote sensing image processing, this part of vegetation was identified as water area rather than vegetation area. Therefore, the vegetation area shows a fluctuating increasing trend.

### 3.3. Variation in Plant Diversity

The plant diversity indices in flooding area and groundwater elevated area of the terminal lake during the whole ecological water transfer period was obtained (Figure 8A,B). There were very few species in 2000 because it had been dried up for a long time before 2000 and later increased significantly. The species number reached 12 in 2005, reached the peak, and, with the change in time and the continuous improvement in the water conditions, plants were gradually replaced with salt-tolerant species (such as *Salicornia europaea* and *Scirpus strobilinus*) and tended to simplify in recent years (Table 2).

**Table 2.** List of important values of species present in quadrats for each year (2000, 2005–2020).

| Plant Species | Life Form of Plant | Important Value | | | | | | | | |
|---|---|---|---|---|---|---|---|---|---|---|
| | | 2000 | 2005 | 2007 | 2009 | 2013 | 2014 | 2015 | 2016 | 2020 |
| *Phragmites australis* | perennial | | 0.10 | 0.17 | 0.25 | 0.29 | 0.72 | 0.66 | 0.92 | 0.82 |
| *Tamarix* sp. | shrub | 0.15 | 0.15 | 0.16 | 0.13 | 0.08 | 0.15 | 0.15 | | 0.01 |
| *Lycium ruthenicum* | shrub | | 0.07 | 0.01 | 0.20 | 0.15 | | | | |
| *Kalidium foliatum* | shrub | | | | | 0.20 | 0.10 | 0.01 | | |
| *Karelinia caspia* | perennial | | 0.16 | 0.08 | | 0.10 | | | | |
| *Launaea polydichotoma* | perennial and annual | | 0.15 | 0.01 | | 0.03 | | | 0.01 | |
| *Apocynum pictum* | perennial | | 0.07 | | | | | | | |
| *Alhagi sparsifolia* | subshrub | | 0.15 | | 0.08 | 0.15 | | | 0.01 | |
| *Scirpus strobolinus* | perennial | | 0.01 | 0.33 | | | | | | |
| *Halocnemum strobilaceum/Halostachys caspica* | shrub | 0.15 | 0.15 | 0.24 | 0.35 | | 0.03 | 0.10 | | 0.01 |
| *Hexinia polydichotoma* | annual herb | | | | | | | 0.08 | 0.07 | 0.16 |

According to the variation characteristics of species importance value in the overflow area in recent 20 years, it is found that the top communities of plant succession are *P. australis* and *Hexinia polydichotoma* (Table 2).

Since the ecological water transportation to Taitema Lake in 2000, the diversity of the surrounding plant communities had increased significantly, and with the progress of ecological water transportation, the benefits of ecological restoration of the vegetation of Taitma Lake had gradually appeared. The increase in diversity was most obvious in the 3 to 5 years after the water transfer. Although the diversity changed in the later period, the diversity index (Simpson index) in each year after the water transfer was greater than the level before the water transfer. It not only protects the survival of the surrounding native vegetation, but also promotes the emergence of *P. australis* and other plants, which was very beneficial to the increase of the biodiversity of the plant community and the restoration of vegetation.

The results of this paper showed: under the artificial water delivery disturbance, the species composition and species diversity decreased significantly in the later stage of 2005–2020. Through the Mann–Kendall trend test, the species richness and Pielou evenness index showed a very significant downward trend ($|Zc| > 2.58$, $p < 0.01$), so species composition tended to simplify gradually in the later stage. Many studies show that, after vegetation is disturbed, the species diversity increases first, and, as the disturbance continues, the plant diversity decreases [6,32]. This is consistent with the research results of this article. In addition, the manifestation of ecological benefits was reflected in the plant diversity indicators in the overflow area and groundwater elevated area. The plant species richness in the overflow area were significantly higher than those in the groundwater elevated area.

From Figure 9, we can also see that the plant diversity index showed a lower level from 2008 to 2010. The main reason is that 2008 and 2009 were dry years, the ecological water did not reach Taitema Lake and the plant diversity also changed with the surrounding water conditions.

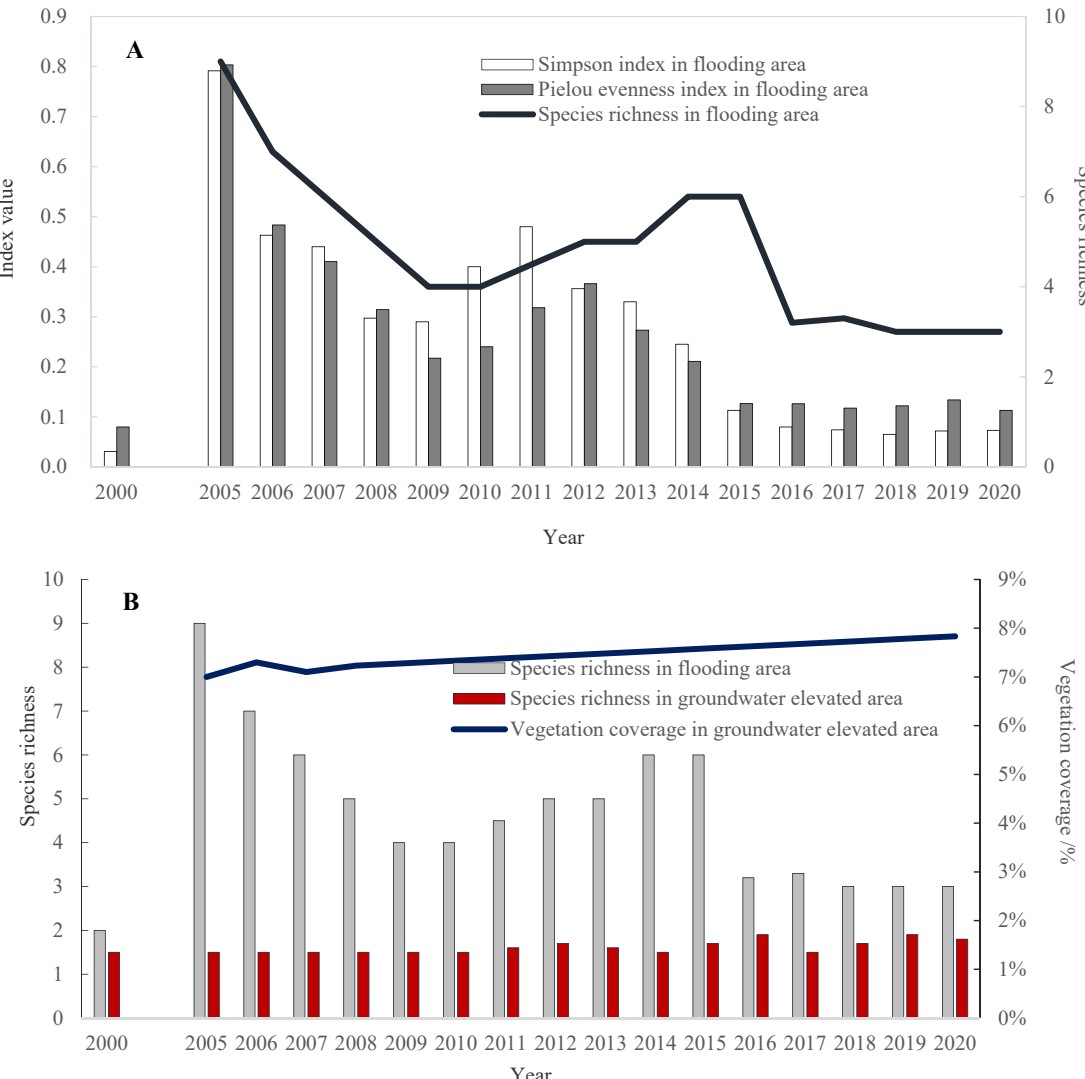

**Figure 9.** Plant diversity indices in flooding area (**A**) and species richness in flooding area and in groundwater elevated area (**B**).

With the initiation of the ecological water conveyance project in 2000, the growth of the surrounding vegetation has improved, vegetation coverage area were expanding and area of sandy land decreased significantly. The species diversity index in the study area

showed a significant increasing trend in the early stage of ecological water conveyance, but showed a very significant downward trend from 2005 to 2020, indicating that the species composition tended to be simplified in the later stage.

Before water conveyance (2000), the vegetation around the lake is a salt firewood shrub community, a mixed community of herbs and shrubs from 2002 to 2009, and a herbaceous community after 2011. In the process of ecological water conveyance in recent 20 years, the vegetation in the overflow area of Taitema Lake has developed towards the salinized Lowland Meadow dominated by *P. australis*, so the species composition is relatively simple. The distribution characteristics of plant community types around the lake are as follows: with the decrease of water, the plant species decrease. In the area of rising groundwater level, woody plants are more than herbaceous plants, accounting for more than 80% of the whole plant species. The occurrence of this phenomenon is a special case under extremely harsh ecological conditions. It can be explained from the perspective of plant ecology that the more severe the habitat of vegetation is, the more obvious the lignification phenomenon is, because shrubs can tolerate the huge fluctuations of the environment better than herbs.

*3.4. Intra-Annual Variation in Lake Area*

Figure 10 showed the intra-annual variation in the water area of the terminal lake. From Figure 10, we can see that the lake area changes significantly within a year. Whether in 2017 or 2018, the lake area showed a downward trend before autumn (August–September). In 2017, from 332 square kilometers in spring to 90 square kilometers in autumn, and from 593 square kilometers in spring to 102 square kilometers in 2018.

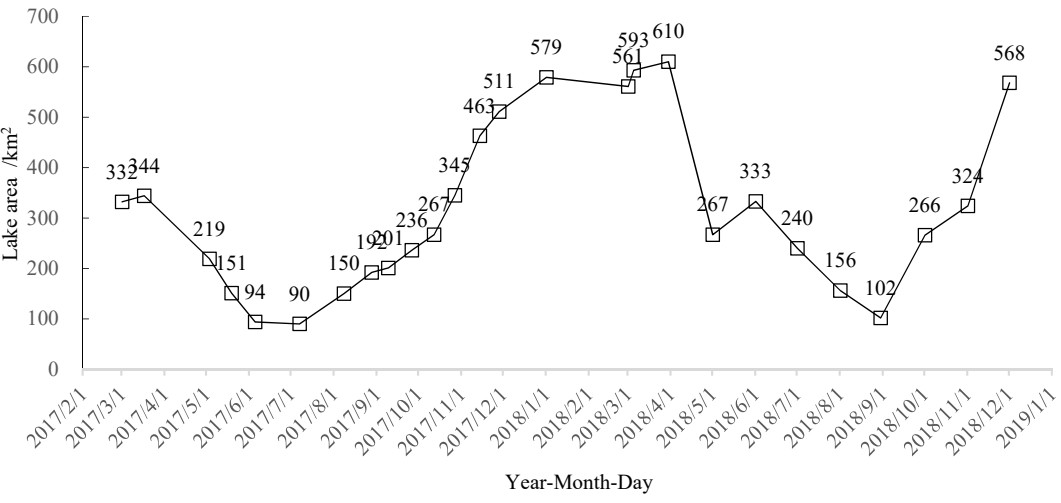

**Figure 10.** Variation curve of water surface area of intra-annual of Taitema Lake.

Although the area of Taitema Lake decreases rapidly within a year, water may enter the lake at the end of the year, and the lake area will increase again at the end of the year. There is an obvious correlation between the lake area and the time of ecological water transfer. When the water transfer ends, the lake continues to decrease, while when the water transfer begins, the lake area increases sharply.

*3.5. Suitable Area for Lake Taitema*

Taitema Lake plays an important role in the downstream ecology [33]. Taitema Lake is located between the two deserts and at the junction of national highway 218 and national highway 315. It is also an important habitat and species gene bank for aquatic animals and plants in the lower reaches of the Tarim River. It is of great strategic significance to maintain a certain scale in both wet and dry years [33]. However, the restoration of Lake Taitema still faces the following problems: with the governance in recent decades, the area of Taitema Lake reached 511 km$^2$ in 2017, the effect of water surface restoration is remarkable. The

large lake area also means that the lake evaporation is large. Too many water resources saved in the upstream are wasted in the lake evaporation. If the water supply is not carried out regularly, the lake area will decrease rapidly. Even in different seasons within a year, there will be significant differences in lake area (as shown in Figure 10). As a result, the growth of coastal vegetation is limited and the distribution range is reduced, resulting in the decline of ecological protection effect. While maintaining the surface area of Taitema Lake, it is bound to have a serious impact on the ecological and agricultural water consumption in the upstream. The total length of the whole Tarim River Basin is 1321 km, and the annual evaporation potential is mostly 1800–2900 mm, and the precipitation is only 18.6–50 mm. In the process of water discharge, it is bound to produce huge losses due to natural evaporation and leakage, so the ecological water cannot be used efficiently, especially during the dry year, it will aggravate the contradiction of water resources, so there is a situation of "more money and low efficiency".

Continuous water delivery was conducive to the restoration of ecological environment, but an excessively large water area was not conducive to the restoration of desert vegetation and the maintenance of diversity. Additionally, in extremely arid areas, water resources were very precious and limited. Therefore, according to the fitting formula of the amount of water entering the lake and the area of Taitema Lake and the changes in the actual lake area of Taitema Lake, the impact of the changes in the lake area on the surrounding organisms and the environment was also considered. It was considered appropriate to control the area of Taitema Lake, about 180 km$^2$ (this area is the average of the annual maximum area in recent 20 years). This would not only enable the rational use of water resources in the Tarim River Basin, reducing the pressure on water transportation in the upper and middle reaches, but also allow Taitema Lake to maximize the local ecological value and maintain the results of water transportation.

## 4. Conclusions

(1) With the progress of ecological water conveyance and the increase of lake area, the groundwater level has increased significantly compared with that before water conveyance. By establishing the relationship, there is a very significant negative correlation between groundwater depth and lake area (the correlation coefficients are −0.79 and −0.70, respectively).

(2) With the initiation of the ecological water conveyance project in 2000, the growth of the surrounding vegetation has improved, vegetation coverage areas were expanding and the area of sandy land decreased significantly. The species diversity index in the study area showed a significant increasing trend in the early stage of ecological water conveyance, but showed a very significant downward trend from 2005 to 2020, indicating that the species composition tended to be simplified in the later stage.

(3) The area of Taitema Lake has changed significantly throughout the year, but the lake water has not disappeared, and the lake area will increase again at the end of the year. There is an obvious correlation between the lake area and the time of ecological water transfer. When the water transfer ends, the lake continues to decrease, while when the water transfer begins, the lake area increases sharply.

**Author Contributions:** Conceptualization and methodology, H.X.; writing—original draft preparation, X.Z.; supervision and data analysis, P.Z.; project administration and funding, H.X.; data processing, P.Z. All authors have read and agreed to the published version of the manuscript.

**Funding:** Project of Construction Management Bureau of Land Development and Consolidation in the Xinjiang Uygur Autonomous Region—A Special Research on Ecological Problems and Ecological Restoration in Tarim River Basin, Xinjiang (JTZB(2021-034 (03)); West Light Foundation of the Chinese Academy of Sciences (Y734341).

**Conflicts of Interest:** The authors declare no conflict of interest.

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
