# Peer review of "Changes of Lake Area, Groundwater Level and Vegetation under the Influence of Ecological Water Conveyance—A Case Study of the Tail Lake of Tarim River in China"

_water, doi:10.3390/w14071026_

Round 1
Reviewer 1 Report
The focus of the article is interesting and it is obvious that the authors did a good study. However in order to be published it needs to deal with the following:
1.The introduction does explain much about the international relevance of this study. It just give specifics about the local situation but considering that the topic is important at international level it does not put the study in an international context. Mainly it does not point out the relevance of the study for the international scientific community.
2.The “Ecological Water Conveyance Project”(EWCP) is not presented. It would be relevant for the readers to know what were the measures implemented in the context of the study
3.Not enough references. The subject is broadly covered in the scientific literature but that literature is not included.
4.Conclusions are to simplistic. As a reader I would be interested into more detailed conclusions about the relevance of the findings for the international scientific community and also about recommendations based on the experience of the authors.
5.The structure of the paper needs to be improved. The paper is very detailed when it comes to methodology and data used but to much general in the introduction and Conclusion sections.
6.Section 2.1. Overview of the Research Area does not fit properly in the Material and Methods section
7.Figure 1 needs refining (the general maps is overlapping with the detailed map), please insert a border to separate them
8.A map showing the distribution of sample plots would be useful
9.Figure 3 needs to be better explained. There are actually 2 figures
Author Response
- The introduction does explain much about the international relevance of this study. It just give specifics about the local situation but considering that the topic is important at international level it does not put the study in an international context. Mainly it does not point out the relevance of the study for the international scientific community.
Reply: It has been revised as required. Please see the revised draft.
- The “Ecological Water Conveyance Project”(EWCP) is not presented. It would be relevant for the readers to know what were the measures implemented in the context of the study.
Reply: China has carried out this project for more than 20 years, and many related introductions about it have also been published internationally. Since 2000, the ecological water conveyance project has transferred the water from a lake to the area of the lower reaches of the Tarim River—Taitema lake. Namely, Tarim River Basin transfers ecological water to Taitema Lake, the tail lake of Tarim River Basin, through water resource conservation and management of the whole basin and water transfer from Bosten Lake.
- Not enough references. The subject is broadly covered in the scientific literature but that literature is not included.
Reply: It has been revised as required. Please see the revised draft.
- Conclusions are to simplistic. As a reader I would be interested into more detailed conclusions about the relevance of the findings for the international scientific community and also about recommendations based on the experience of the authors.
Reply: The conclusion part of the revised draft is the final conclusion drawn after discussion. It is mainly aimed at the result part, one-to-one discussion and one-to-one summary. Therefore, the discussion part and the conclusion part cannot be separated. I wonder if you are satisfied with this modification? If not, we will separate them and revise them again.
- The structure of the paper needs to be improved. The paper is very detailed when it comes to methodology and data used but to much general in the introduction and Conclusion sections.
Reply: It has been revised as required. Please see the revised draft.
- Section 2.1. Overview of the Research Area does not fit properly in the Material and Methods section
Reply: As for the introduction of the study area, the revised draft has revised it, deleted irrelevant contents, and added contents such as soil characteristics.
- Figure 1 needs refining (the general maps is overlapping with the detailed map), please insert a border to separate them图1需要细化(总图与明细图重叠),请插入边框将其分开
Reply: Figure 1 has been revised following the above.
- A map showing the distribution of sample plots would be useful
Reply: Figure 2 (showing the distribution of sample plots) has been added.
- Figure 3 needs to be better explained. There are actually 2 figures
Reply: Figure 3B has been separated from figure 3a, and Figure 3B has been combined with the following contents related to groundwater. See the revised draft for details.

Reviewer 2 Report
A brief summary
A review of the manuscript entitled: „ Changes of lake area, groundwater level and vegetation under the influence of ecological water conveyance - a case study of the tail lake of Tarim River in China”is very interesting and promising. In general, the manuscript is well written but I think that the authors need to make some insightful improvements. Based on this general evaluation and the specific comments, reported below, I recommend a major revisions of the manuscript and re-write and re-organization before it will be acceptable for publication. I have few specific comments, which might improve the manuscript.
Introduction
The Authors use the term ‘tail lake’ which is not known in the scientific literature. I did not find this term in the databases: Scopus, Web of Sciences. The term ‘tail lake’ should be defined and described in the introduction.
- 84 - the term „vegetation evolution” sounds wrong. What does it mean? Should be replaced with changes in vegetation or dynamic vegetation
Materials and Methods
- 224 – Simpson index equation is given according to (2) = Abudumijiti (2015). This publication does not describe Simpson index. Original publication: Simpson, E H. 1949. Measurement of diversity. Nature 163, p. 688.
- 226 – chinese font?
- 226 – Shannon-Wiener equation is given according to (4) = Brancalion et al. 2014. This publication does not use the Shannon-Wiener index. Original publication: Shannon, Cloud E. 1948. A mathematical theory of communication. The Bell System Technical Journal 27: 379–423, 623–656.
Results
No basic statistical analyses. Would be nice to see statistical results and statistical tests eg. comparsions of diversity indices (Simpson index, Shannon-Wiener index, Pielou evenness index) over the years or flooding area and in groundwater elevated area.
Specific comment
Current species names should be provided, according to The Plant List (http://www.theplantlist.org/):
- Hexinia polydichotoma = Launaea polydichotoma (Ostenf.) Amin ex N.Kilian is an accepted name;
- Poacynum hendersonii = Apocynum pictum Schrenk is an accepted name
Check the plant names in the whole text.
Table 2 - typo in plant name – ‘Scirpus strobilinus’ replace with Scirpus strobolinus
Table 2 - Hexinia polydichotoma = Launaea polydichotoma classified as perennial and annual herb plant, please, clarify.
Author Response
Reviewer 2:
A review of the manuscript entitled: „ Changes of lake area, groundwater level and vegetation under the influence of ecological water conveyance - a case study of the tail lake of Tarim River in China”is very interesting and promising. In general, the manuscript is well written but I think that the authors need to make some insightful improvements. Based on this general evaluation and the specific comments, reported below, I recommend a major revisions of the manuscript and re-write and re-organization before it will be acceptable for publication. I have few specific comments, which might improve the manuscript.
Introduction
The Authors use the term ‘tail lake’ which is not known in the scientific literature. I did not find this term in the databases: Scopus, Web of Sciences. The term ‘tail lake’ should be defined and described in the introduction.
Reply:By consulting the relevant literature, one part uses "tail lake" and the other part uses "terminal lake", as shown below. I very much hope that experts can help me choose one of them two definitions and I will revise it.
84 - the term „vegetation evolution” sounds wrong. What does it mean? Should be replaced with changes in vegetation or dynamic vegetation
Reply: Many thanks for your advice. “vegetation evolution” has been changed to “dynamic vegetation”.
Materials and Methods
224 – Simpson index equation is given according to (2) = Abudumijiti (2015). This publication does not describe Simpson index. Original publication: Simpson, E H. 1949. Measurement of diversity. Nature 163, p. 688.
Reply: Many thanks for your advice. This paper has cite the reference of “ E. H. Simpson, “Measurement of Diversity,” Nature, Vol. 163, 1949, pp. 688-690.”
226 – chinese font?
Reply: We are very sorry for the emergence of Chinese. Now it has been removed.
226 – Shannon-Wiener equation is given according to (4) = Brancalion et al. 2014. This publication does not use the Shannon-Wiener index. Original publication: Shannon, Cloud E. 1948. A mathematical theory of communication. The Bell System Technical Journal 27: 379–423, 623–656.
Reply: Many thanks for your advice. This paper has cite the reference of “Shannon, C.E. The mathematical theory of communication. Bell Syst. Tech. J. 1948, 27, 379–423..”
Results
No basic statistical analyses. Would be nice to see statistical results and statistical tests eg. comparsions of diversity indices (Simpson index, Shannon-Wiener index, Pielou evenness index) over the years or flooding area and in groundwater elevated area.
Reply: In factors in each year have an average value and standard deviation, but the vegetation coverage does not. The change characteristics of several plant indicators in Figure 8 in recent 20 years are displayed (at the same time, some indicators are also divided into overflow area and non-overflow area). There are many indicators and long year series. Therefore, a figure is divided into two figures a and B. some indicators have standard deviation, while some indicators (such as vegetation coverage) have large numerical differences in different places due to the large amount of data, It has not been fully determined yet. In order to ensure the unity of Fig. 8a and Fig. 8b, no standard deviation is marked. The trend change test of these indicators has appeared in the research methods and results and analysis.
Figure x. Species richness (A) and Pielou evenness index (B) of flooding area and groundwater elevated area of the terminal lake of Tarim River.
Specific comment
Current species names should be provided, according to The Plant List (http://www.theplantlist.org/):
- Hexinia polydichotoma = Launaea polydichotoma (Ostenf.) Amin ex N.Kilian is an accepted name;
Reply: Thank you for your advice. This has been corrected.
- Poacynum hendersonii = Apocynum pictum Schrenk is an accepted name
Check the plant names in the whole text.
Reply: Thank you for your advice. This has been corrected.
Table 2 - typo in plant name – ‘Scirpus strobilinus’ replace with Scirpus strobolinus
Reply: Thank you for your advice. This has been corrected.
Table 2 - Hexinia polydichotoma = Launaea polydichotoma classified as perennial and annual herb plant, please, clarify.
Reply: Thank you for your advice. This has been corrected.
By reviewing http://www.theplantlist.org/ and flora of China, Launaea polydichotoma is perennial and annual herb plant.

Reviewer 3 Report
This article focuses on an interesting and relevant problem. At the end of the introduction, a question arises: How large should the appropriate water area of the terminal lake be? Is it absolutely certain that the bigger the better"? Readers expect the present paper to answer these questions. However, the answer to these questions is not found at the end of the mauscript. This research will be the more relevant the better this question is answered. And the work has to be reorganized to answer this question. And so, the work will be publishable.
The references have to be increased. There are plenty of publications about this ecosystem that can and should be cited (e.g. https://www.nature.com/articles/s41598-021-96742-5
https://doi.org/10.1016/j.envres.2019.109009
https://doi.org/10.1016/j.catena.2021.105725
https://doi.org/10.3390/rs14040832) and many more. The discussion contains almost no citations to support or refute the authors' conclusions. Perhaps it would be better to combine the results and the discussion in a single item.
The authors can find more suggestions and observations in the manuscript.

Author Response
Reviewer 3:
This article focuses on an interesting and relevant problem. At the end of the introduction, a question arises: How large should the appropriate water area of the terminal lake be? Is it absolutely certain that the bigger the better"? Readers expect the present paper to answer these questions. However, the answer to these questions is not found at the end of the manuscript. This research will be the more relevant the better this question is answered. And the work has to be reorganized to answer this question. And so, the work will be publishable.
Reply: The answers to the above questions have been presented in the "discussion and conclusion" in the revised draft. After long-term discussion and consultation with the old experts of Taitema Lake research, we finally think that according to the area in recent 10-20 years, we can calculate an arithmetic average value, which should be the appropriate area.
- Abstract: “To examine the variation in water and vegetation coverage areas, the groundwater level 11 and plant diversity in the terminal lake of the Tarim River, northwest China, both the monitoring 12 data of a field survey consisting of surface samples and remote sensing data for 20 years (2000–2019) 13 were analyzed by using field survey and indoor remote sensing methods.”This sentence is too long and a bit unclear.
Reply: The following expressions are formed: “To study the changes of water and vegetation coverage, groundwater level and plant diversity of lakes at the end of Tarim River in Northwest China, the changes of various indicators in more than 20 years (2000-2019) were analyzed through field investigation and indoor remote sensing methods.”
- Also with consequences for nutrients internal load in the lakes, for fish species and, enventualy, fisheries, for migratory birds and others. These aspects should be mentioned herein.
Reply: Thank you for your advice. The "impact of nutrients in lakes, fish and ultimately fisheries, migratory birds and other organisms" has been added to the corresponding position as follow:
- which seriously threaten the sustainable development of mankind. Is it the adequate word?
Reply: “mankind”has been changed to “Human sustainable development”.
- It would be interesting to see a very succinct of this project in the introduction... the water became available after changing the irrigation practices in this basin? The water for this river was transfered from another basin?
Reply: Due to space constraints, the EWCP project will not be described in detail in the manuscript. China has carried out this project for more than 20 years, and many related introductions about it have also been published internationally. Since 2000, the ecological water conveyance project has transferred the water from a lake to the area of the lower reaches of the Tarim River—Taitema lake.
- Please clarify... It seems that this sentence is out of context
Reply: yes, this sentence is out of context, so was deleted. As for the introduction of the study area, the revised draft has revised it, deleted irrelevant contents, and added contents such as soil characteristics.
- The choice of sampling plots was influenced by the road network proximity ...maybe is more concise and clear to the readers.
Reply: Regarding ”Considering the accessibility of the sampling plots, the layout of the sampling plots, which is illustrated in Table1, was influenced by the road network in the area and accessibility of various sections.” It has been expressed as ” The choice of sampling plots was influenced by the road network proximity, the final sample plots in Table 1 were determined after multi-directional layout and multiple screening in the process of sample plots selection.”
- Repetitive: Erase please
Reply: The above repeated statements have been deleted.
- Figure 2 shows the composition of vegetation area of different coverage gradients. Vegetation coverage was divided into 6 different gradients, namely,
Reply: Many thanks for the advice. Regarding” Figure 2 shows the composition of vegetation area of different coverage gradi-ents, vegetation coverage was divided into 6 different gradients, namely, 1) Ssandy land (vegetation coverage < 103%).” We have expressed it as
” Figure 2 shows the composition of vegetation area of different coverage gradients. Vegetation coverage was divided into 6 different gradients, namely, 1) sandy land (vegetation coverage < 10%).”
- were used
Reply: Many thanks for the advice. Regarding” level are used”, it has been expressed as ”level were used”
Reply: I'm very sorry for the emergence of Chinese. Now it has been removed.
- This graphic must be improved.
Reply: The figure on the left has been modified, as shown in the figure on the right. Is this ok? If not, I'll revise it again.
- Variation of groundwater level monitored by two wells in Taitema Lake. this data should be pooled together with a line showing lake area variation
Reply: Now the groundwater map in Figure 4 and figure 3 has been combined. See the revised draft for details.
- Why?
Reply: The study area is in the hinterland of Taklimakan Desert. Two ecological monitoring wells are set up near the tail lake to monitor the groundwater level. Through years of field investigation and monitoring, it is found that in wet years, there is more water entering the tail Lake in the upstream, part of which forms the lake surface, part of which enters the groundwater and part of which is evaporated. When the amount of water entering the lake in a certain year is small, it indicates that this year is a dry year and has little supplementary effect on groundwater.
- What is the meaning of "important values" is it an index combination? please clarify
Reply: we added follows:
Determination of important value:
Important value can reflect the dominant species in the community and objectively express the status and role of each species in the community. Its calculation method (Hao et al.,2016) is as follows:
Important value = (relative density relative coverage relative frequency) / 3 (2)
Relative density = number of plants of a species / sum of the number of plants of all species ×100%
Relative coverage = coverage of a species / coverage of all species and ×100%
Relative frequency = frequency of a species / sum of frequencies of all species ×100%
We account for the number of species, calculated the species frequency and the Pielou evenness index in each quadrant.
- What does the line mean?
Reply: The line means the vegetation coverage in groundwater elevated area. The change of this line over time is an indicator to be displayed on the vertical coordinate on the right.
Disscussion
- Maybe results and discussion should coming together...
Reply: Yes, the results and discussion have been combined.
- Why?
Reply: More than 50 years ago, people in the Tarim Basin did not use water as strongly as they do now. There is often water in the tail lake of the Tarim River - taitema lake, which has always been a transit station and habitat for many migratory birds flying from China to Siberia. However, in the past 50 years, the environment of Taitema lake has become worse and worse, accompanied by the extinction of animals and plants. Therefore, under such a good opportunity (the state attaches importance to ecological construction), the tail lake should maintain a certain water surface (whether in wet season or dry season) to create some contributions to the ecology of migratory birds and the ecology of the earth.
- Why? cite other authors to support this sentence. Are other research in line with these results?
Reply: Yes, other researches may be with the same results. Such as, Fan Zili et al.(2014) on the discussing on suitable area of Taitma Lake, Fu Aihong et al.(2021) on the effects of “EWCP” (ecological water conversion project) to riparian forest ecosystem in the Tarim River Basin, etc.
- why is graphic was not presented in fig.4?
Reply: The purpose of the above figure related to groundwater is to express that in recent 20 years, with the increase of the lake area and the formation of a certain scale, the groundwater level near the study area has also been improved, which can provide water for the vegetation in the groundwater level rise area (non-overflow area), that is, to emphasize that the groundwater level has become better. The purpose of the following diagram related to groundwater is to express that the response process of groundwater level in the study area is too slow and has a strong lag. Therefore, the purpose is different, and the two cannot be combined.
- is it a citations?
- 参考文献
The references have to be increased. There are plenty of publications about this ecosystem that can and should be cited (e.g. https://www.nature.com/articles/s41598-021-96742-5
https://doi.org/10.1016/j.envres.2019.109009
https://doi.org/10.1016/j.catena.2021.105725
https://doi.org/10.3390/rs14040832) and many more. The discussion contains almost no citations to support or refute the authors' conclusions. Perhaps it would be better to combine the results and the discussion in a single item.
The authors can find more suggestions and observations in the manuscript.

Round 2
Reviewer 2 Report
Dear Authors,
In my opinion term "tail lake" is the most appropriate. Perhaps it would be useful to explain the term "tail lake" for the readers. Rather, it is not known to the international community.
Author Response
Reviewer 2:
In my opinion term "tail lake" is the most appropriate. Perhaps it would be useful to explain the term "tail lake" for the readers. Rather, it is not known to the international community.
Reply: Thank you very much for your advice.

Reviewer 3 Report
The article still needs to be a bit more organized before it can finally be published. My suggestions are in the manuscript. I suggested to the authors to join the results with the discussion and what they did was to join the discussion with the conclusions. However, I think the paper is ready to be accepted after this round.

Author Response
Reviewer 3:
The article still needs to be a bit more organized before it can finally be published. My suggestions are in the manuscript. I suggested to the authors to join the results with the discussion and what they did was to join the discussion with the conclusions. However, I think the paper is ready to be accepted after this round.
1、“Also with consequences for nutrients internal load in the lakes, for fish species and, eventually, fisheries, for migratory birds and others.”should be cited.
Reply: Yes, the above text is quoted from the following literature:
2、“on the dynamic vegetation?” or“on the vegetation dynamic?”
Reply: We have changed the “dynamic vegetation” to “dynamic vegetation”.
3、Put a A and a B in graphics 7.
Reply: We have put A and B in graphics 7.
4、Please review the figures. Figure 5B is missing.
Reply: For the description part of Fig. 5, the original figure 4B should be changed to the current figure 5B. It is now complete.
5、Please, Join the results with the discussion (NOT DISCUSSIONS) and separate the conclusions.
Reply: We have combined the results with the discussion and separated the conclusions. Please see the revised draft for details
